# Effects of integrated economic and health interventions with women's groups on health-related knowledge, behaviours and outcomes in low-income and middle-income countries: a systematic review protocol

Sapna Desai ,[1] Kala M Mehta,[2] Roopal Jyoti Singh,[1] Allie K Westley,[3] Osasuyi Dirisu,[4] Connie Wong,[5] Thomas De Hoop,[6] Gary L Darmstadt [7]

For numbered affiliations see end of article.

**Correspondence to**
Dr Gary L Darmstadt;
gdarmsta@stanford.edu

## ABSTRACT

**Introduction** Economic groups, such as microfinance or self-help groups are widely implemented in low-income and middle-income countries (LMICs). Women's groups are voluntary groups, which aim to improve the well-being of members through activities, such as joint savings, credit, livelihoods development and/or health activities. Health interventions are increasingly added on to existing women's economic groups as a public health intervention for women and their families. Here, we present the protocol for a mixed-methods systematic review we will conduct of the evidence on integrated economic and health interventions on women's groups to assess whether and how they improve health-related knowledge, behaviour and outcomes in LMICs.

**Methods and analysis** We will search seven electronic databases for published literature, along with manual searches and consultation. The review will include (1) randomised trials and non-randomised quasiexperimental studies of intervention effects of integrated economic and health interventions delivered through women's groups in LMICs, and (2) sibling studies that examine factors related to intervention content, context, implementation processes and costs. We will appraise risk of bias and study quality using standard tools. High and moderate quality studies will be grouped by health domain and synthesised without meta-analysis. Qualitative evidence will be thematically synthesised and integrated into the quantitative synthesis using a matrix approach.

**Ethics and dissemination** This protocol was reviewed and deemed exempt by the institutional review board at the American Institutes for Research. Findings will be shared through peer-reviewed publication and disseminated with programme implementers and policymakers engaged with women's groups.

**PROSPERO registration number** CRD42020199998.

## Strengths and limitations of this study

► We outline the protocol for a systematic review of current global literature exploring a highly important method of health education delivery for women.

► This convergent mixed-methods review will use robust methods to integrate quantitative evidence on intervention effectiveness with qualitative studies to provide programme and policy-relevant evidence on the effects of integrated women's groups interventions on health outcomes.

► The review will include sibling studies, such as process evaluations identified through a comprehensive search of published literature and will summarise intervention content, context, implementation processes, facilitating factors and barriers, and costs.

► The review will include a comprehensive risk-of-bias assessment to critically appraise study quality.

► The review is limited by lack of meta-analysis, given heterogeneity of reported outcomes.

## INTRODUCTION

Women's groups come together in many settings in low-income and middle-income countries (LMICs) to improve women's opportunities for livelihoods and financial security.[1] Women's self-help groups (SHGs) and collectives have developed and spread throughout India for several decades; the National Rural Livelihoods Mission has reached over 70 million households with microfinance and livelihoods interventions through SHGs since 2011.[2] Similarly, Village Savings and Loan Associations and other savings groups have operated in many parts of Africa for over three decades.[3] Economic women's groups typically focus on savings, access to credit and livelihoods to improve financial inclusion and economic outcomes. A recent systematic review found that livelihoods groups have small, but positive effects

on consumption and moderate effects on savings, but did not find effects on income, asset ownership or labour force participation.[1] The review suggests that livelihoods groups may have more impact when combined with investments in human capital, including health. Previous reports of the effect of combined microfinance and health interventions, though not all group-based, suggest some evidence of improvements in health behaviours among participants.[4 5]

Governments and other stakeholders increasingly support integrating (or 'layering') additional activities onto livelihoods-based groups, most commonly with health interventions. Combined economic and health interventions implemented through groups have been evaluated for impacts on a range of health outcomes, including violence against women,[6 7] health-promoting behaviours[8] and malnutrition among women and children.[9] Despite growing policy interest, there has been no systematic review of the evidence on combined economic and health interventions through women's groups to inform ongoing and future programmes in low-income and middle-income settings. We build on previous reviews by (1) focusing on the combined effect of group-based economic and health programmes, as distinct from previous reviews that examined the health impacts of joining a microfinance-based group with or without layering[4] or non-group-based microfinance interventions[5] and (2) including studies across all LMICs rather than a specific region.[10 11]

This paper presents the protocol, following Preferred Reporting Items for Systematic Review and Meta-analysis Protocols guidelines,[10] for a review we will conduct, which aims to address this gap through synthesising quantitative and qualitative evidence on the health effects of combined group-based economic and health interventions and implementation factors that contributed to achieving intended outcomes. We aim to answer two primary questions: (1) What is the effect of combined economic and health, nutrition and/or sanitation interventions delivered through women's groups on health outcomes among women and children in LMICs, compared with single-purpose, non-layered (ie, only economic or only health) interventions with women's groups or to no group intervention, and (2) what factors related to intervention content, context and implementation processes are enablers or barriers to achieving health outcomes?

## METHODS
### Study design
#### Study inclusion criteria
This review is a mixed-methods convergent review in which synthesis of quantitative effects and qualitative evidence for factors in intervention implementation will be combined within the review.[12] Our review will include all trials—both randomised controlled trials and non-randomised, and quasiexperimental studies—on integrated economic and health interventions through

groups. Non-randomised studies of interventions (NRSI) include stronger and weaker designs.[13] Criteria for inclusion of NRSI will be the ability to address selection bias either through the inclusion of a baseline measurement of the outcome of interest or other relevant confounding factors, or through intervention allocation rules that enable the use of regression discontinuity designs or natural experiments.[14 15] Additional inclusion criteria include that the intervention was implemented in an LMIC according to World Bank classification[16] and the study was published between 1 January 2000 and 31 May 2020 in the peer-reviewed or grey literature. We will also include studies, which are linked to the included randomised and non-randomised trials, including process evaluations, qualitative studies and cost or cost-effectiveness studies, that is, 'trial sibling studies'.[17]

#### Study exclusion criteria
Studies with the following characteristics will be excluded: (1) No qualitative or quantitative data were collected to evaluate the effect of an intervention, for example, the study only provides observational descriptions of groups; (2) the treatment group only implements an economic or health intervention, that is, the intervention is not an integrated approach; (3) no comparison group or baseline measurement was included; (4) an independent study, such as a process evaluation, qualitative study and cost or cost-effectiveness study on women's groups was not linked to a randomised or non-randomised impact evaluation (non-sibling studies); and (5) the study was published prior to 1 January 2000.

### Participants
#### Participant inclusion criteria
For this review, we define an eligible women's group as a group that was constituted for women to regularly exchange information, support or goods (eg, savings) and plan individual and/or collective action. Mixed-sex groups will be included if there were 51% or more women or authors note the group was composed mostly of women. We will include closed groups and groups constituted for women that were also open to others to attend meetings. We define adult women to be 18 years of age or more.

#### Participant exclusion criteria
Our review will exclude men's only groups, family groups, select committees of community representatives, professional groups and political associations. We will also exclude adolescent girls' groups, as a recent systematic review examines these groups and the role of integrated interventions compared with singular approaches,[18] as well as groups that did not have a joint purpose or objective to meet regularly, such as short-term training sessions.

### Types of interventions
#### Intervention inclusion criteria
The review will focus on integrated economic and health/sanitation/nutrition interventions delivered through

women's groups. Groups may be pre-existing in the study areas or may have been set up as part of the group intervention. Interventions will include combinations of health and/or nutrition and/or sanitation components with an economic intervention aimed to improve financial inclusion, income, consumption or livelihoods, such as microfinance, entrepreneurial skills training or collective enterprise development.

### Intervention exclusion criteria

We will not examine economic women's groups that did not have a health, sanitation or nutrition intervention component (defined below), or women's groups that solely had health interventions, without economic interventions, such as mothers' groups. If the intervention was not described in sufficient detail to categorise the type of intervention, it will not be included.

### Comparisons

Comparisons will include groups with non-intervention controls and 'usual' practice. We will include evaluations that compared the integrated intervention to either no group or to a singular model (economic or health). For example, groups with an economic component, but no health, sanitation, or nutrition intervention can serve as the comparison group.

### Outcomes

Condition or domain being studied will include any health, nutrition or sanitation-related knowledge, behaviours and outcomes across all domains consistent with the WHO definition of health as 'a state of complete physical, mental and social well-being'.[19] Domains will include (but are not limited to): maternal, newborn and child health; sexual and reproductive health; nutrition; infectious and vector-borne disease; non-communicable disease; mental health; violence against women; water, sanitation and hygiene; health services utilisation; and health expenditure. The domain of maternal health, for example, may include both objectively measured health outcomes, such as maternal mortality and anaemia, along with behaviours during the perinatal period and knowledge of postnatal care. Infectious and vector-borne disease may include outcomes related to vector density or incidence of malaria, along with behaviours, such as sleeping under a bednet or seeking treatment for fever and knowledge of symptoms of malaria or dengue.

### Information sources
#### Electronic databases

We will search the following seven electronic research databases: PubMed, Scopus, Embase, Web of Science, PsycINFO, Social Sciences Citation Index and EconLit.

#### Other sources

Studies will also be obtained for the same time period from reference lists of included studies, reference lists of systematic reviews on the effects of women's groups, the 3ie Database of Impact Evaluations and Evidence Consortium on Women's Groups repository, and Google Scholar. In addition, we will contact study authors to identify process evaluations or qualitative studies conducted in conjunction with an impact evaluation and consult with experts familiar with interventions in Francophone Africa and Latin/South America to recommend inclusion of French or Spanish language articles. We will also contact study authors of published process evaluations or qualitative studies to identify linked, unpublished impact evaluations. Finally, we will consult with experts in this area, both researchers and practitioners, to review our list of included studies to identify any interventions that may have been excluded.

### Search strategy

The search strategy for PubMed is in online supplemental appendix 1. This strategy will be adapted to the other electronic databases by the library scientist on our team (CW), with any modifications reported in the review manuscript.

### Data management

The titles and abstracts retrieved by electronic searches will be exported to a systematic review application called Covidence. Reference management for full text will be conducted using EndNote.

### Selection strategy

Two reviewers will independently screen abstracts and titles. The reviewers will not be blind to the author or journal information. The abstracts of papers that are in a language other than English will be translated using Google Translate. If considered eligible or eligibility is unclear, professional translation of the full paper will be undertaken. The full texts of articles will be obtained for all potentially eligible studies for further examination. For all excluded articles, the primary reason for exclusion will be recorded and documented in the excluded studies table. Discrepancies between the two review authors regarding study eligibility will be resolved by discussion and consensus, and if necessary, by a third reviewer.

### Data collection process

Two investigators will independently collect data from articles selected for full text review using an extraction form developed by the authors and piloted before use. Discrepancies between reviewers regarding data extraction will be resolved by discussion and consensus, and if necessary, by contacting authors.

### Data items

The following information will be extracted into Covidence and an Excel file (see the full list of data extraction items in online supplemental appendix 2): publication details: authors, year and journal; study context and study design; hypothesis tested and theory of change; intervention design; level of community participation; social and behaviour change approach; participant inclusion criteria and demographic characteristics; characteristics

of the intervention, including the duration, intervention strategies, intensity and coverage; outcomes (primary and secondary); data collection and statistical methods, including sample size; findings of process evaluations and cost-effectiveness; source(s) of research funding and potential conflicts of interest; and study limitations in the authors' own words. Attempts will be made to contact the corresponding authors of included studies if there are specific queries regarding the items for extractions that are unavailable in the published manuscript.

## Outcomes and prioritisation

We will extract data on all health outcomes, as detailed above. Primary outcomes, as identified by the authors, will be prioritised in reporting. If authors have not distinguished between primary and secondary outcomes, we will report all outcomes. In addition, we will extract data on non-health outcomes reported by authors, such as on women's empowerment or economic outcomes.

## Risk of bias in individual studies

We will appraise risk of bias in trials using the Cochrane guides for randomised controlled trials (Cochrane ROB-2) and for assessing risk of bias in non-randomised studies of interventions (ROBINS-1).[20 21] We will examine risk of bias for primary outcomes. If a primary outcome is not specified, we will use the outcome for which sample size was determined. Process evaluations will be appraised using an adapted version of an eight-item tool developed by the EPPI-Centre, with additions specific to group-based interventions.[22 23]

## Data synthesis

We will first summarise characteristics of all trials with their risk of bias appraisal, grouped by health domain. Given the anticipated heterogeneity of health outcomes, we do not plan to conduct a meta-analysis. We will synthesise findings from trials with low or moderate risk of bias in tabular form and in the narrative text. We will present forest plots by health domain without a summary measure if possible, and potentially present harvest plots to compare effects by factors related to the health intervention approach. Synthesis methods and results will be presented according to the Synthesis without meta-analysis guidelines.[24]

Findings from process evaluations and other qualitative studies will be analysed separately in a thematic synthesis.[25] We will integrate the synthesis on intervention effects with the qualitative synthesis by juxtaposing findings in a matrix.[26] If consistent programme theories emerge across interventions, we may conduct a synthesis by programme theory.

## Metabiases

We will plot study sample size against effect size to evaluate whether these are skewed and asymmetrical in the presence of publication bias and other biases.[27] In addition, the risk of bias assessment tools will identify whether

studies indicate selective reporting of outcomes, which we will summarise across studies if possible.

## Confidence in cumulative evidence

The overall quality of evidence on outcomes will be presented using the Grades of Recommendation, Assessment, Development and Evaluation criteria.[28] This method will include an evaluation of within-study risk of bias (methodological quality), directness of evidence, heterogeneity, precision of effect estimates and risk of publication bias.

## Patient and public involvement

There will be no patient or public involvement in the conduct or dissemination of the results of this study.

## Ethics and dissemination

Our findings will be submitted for peer-reviewed publication. Deviations from the study protocol will be noted in the manuscript. Findings will be disseminated through conference presentations and shared with both programme implementers and policymakers engaged in interventions with women's groups, as well as on the website of the Evidence Consortium on Women's Groups (https://womensgroupevidence.org/). No primary data collection will be undertaken and the institutional review board at the American Institutes for Research in Washington, D.C., determined that the review was exempt from human subjects review.

## DISCUSSION

This proposed review will contribute to a growing body of literature regarding how different types of women's groups may contribute to improved health outcomes.[4 10 11] The focus on integrated, group-based interventions is novel and also timely, as several countries and organisations are considering scaling these interventions in a variety of geographic contexts. Building a systematically aggregated, critically appraised knowledge base will aid in discerning the up-to-date evidence base regarding the health effects of integrated economic and health interventions. This review will also report on different types of women's groups, building on existing typologies,[29] to advance understanding on how interventions may be integrated. Systematically synthesising the evidence for health effects as well as facilitating factors in creating health benefits will aid understanding of which integrated women's groups interventions will potentially work, for whom, in what context, and with which pre-conditions. Further, the incorporation of a mixed-methods approach and qualitative synthesis of implementation evidence will enable better understanding of how to implement women's group interventions effectively in various contexts.[26]

It is the intention that this review will be published in a peer-reviewed journal, will be presented at relevant global health and development conferences and the learnings

will be disseminated to key stakeholders working with or funding work with women's groups in LMIC contexts. Not only will this work be important in and of itself, it will also shape and define several new research questions that will be critical to the field as it moves forward.

**Author affiliations**
¹Population Council India, New Delhi, Delhi, India
²Department of Epidemiology and Biostatistics, University of California San Francisco, San Francisco, California, USA
³Stanford University, Stanford, California, USA
⁴Population Council Nigeria, Utako, Nigeria
⁵Lane Medical Library, Stanford University School of Medicine, Stanford, California, USA
⁶International Development Division, American Institutes for Research, Washington, DC, USA
⁷Department of Pediatrics, Stanford University School of Medicine, Stanford, California, USA

**Contributors** SD, GLD and TDH conceptualised the review. SD and KMM drafted the protocol, with critical inputs from all authors. CW developed the search methodology. RJS, OD and AKW developed and piloted extraction sheets. All authors reviewed and approved the final protocol.

**Funding** This work was supported by the Bill & Melinda Gates Foundation in support of the Evidence Consortium on Women's Groups, through grant number OPP1201417. The funder provided input to the protocol, but the senior author had the responsibility for the content and the decision to publish the protocol.

**Competing interests** None declared.

**Patient consent for publication** Not required.

**Provenance and peer review** Not commissioned; externally peer-reviewed.

**ORCID iDs**
Sapna Desai http://orcid.org/0000-0003-2596-9726
Gary L Darmstadt http://orcid.org/0000-0002-7522-5824

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
