## [Reviewer comments · BMJ Open]

ARTICLE DETAILS

TITLE (PROVISIONAL)	Effects of integrated economic and health interventions with women's groups on health-related knowledge, behaviours and outcomes in low-and middle-income countries: a systematic review protocol
AUTHORS	Desai, Sapna; Mehta, Kala; Singh, Roopal; Westley, Allie; Dirisu, Osasuyi; Wong, Connie; De Hoop, Thomas; Darmstadt, Gary

VERSION 1 – REVIEW

REVIEWER	Rosenberg, Molly Indiana University School of Public Health-Bloomington, Epidemiology and Biostatistics
REVIEW RETURNED	18-Dec-2020

GENERAL COMMENTS	The authors present a protocol for a systematic review of the health effects of integrated health and economic women's groups in low-and middle-income settings. I am enthusiastic about this review: it is well-conceptualized and will make contributions towards synthesizing our knowledge towards implementing more effective interventions in the future. I note these very few suggestions to incorporate into the protocol to improve it's clarity and completeness:  1. It is unclear to me whether observational studies (cross-sectional or longitudinal) with appropriate control groups and adjustment for confounding would be included or excluded. 2. This sentence, p.8, line 14: "An independent studies on women's groups was not linked to an impact evaluation..." is unclear 3. Given the requirement of an economic component to the interventions, the authors should also consider abstracting articles from an economic journal database (EconLit: https://www.aeaweb.org/econlit/), where at least some microfinance group literature is published.
--

REVIEWER	Flax, Valerie RTI International
REVIEW RETURNED	04-Jan-2021

GENERAL COMMENTS	The review described by the authors is important as funders move forward with support for integrated or layered programming. It would be helpful if the authors could explain in the manuscript how they think their review differs from or adds to the previous two reviews by Orton et al. and Lorenzetti et al. I would also like to offer the following minor comments for the authors' consideration: Intro, page 5 – first research question – It's not clear what the authors mean by "single-purpose interventions". Does that mean only an economic intervention without any layering? The authors may want to consider modifying the wording for clarity.
---

	Page 6, lines 9-12 – Points 3 and 4 here seem contradictory. Does point 3 only refer to randomized studies? Page 6, line 28 – It may be hard to determine what percentage of the participants in a group were women in the case of mixed groups. Not all authors include the exact percentages of female and male group members. They may say “mostly women” or “a majority of the group members were women”. What will you do in such cases? Page 8, other sources – The authors mention earlier in the manuscript that they will search the grey literature, but in the “other sources” section there is no mention of grey literature searches. Will the authors search or contact some of the main NGOs that implement financial inclusion programs – CARE, Catholic Relief Services, Grameen Foundation, etc. – to obtain grey literature on combined/integrated programs?
--	--

VERSION 1 – AUTHOR RESPONSE

Reviewer: 1

Dr. Molly Rosenberg, Indiana University School of Public Health-Bloomington

Comments to the Author:

The authors present a protocol for a systematic review of the health effects of integrated health and economic women’s groups in low- and middle-income settings. I am enthusiastic about this review: it is well-conceptualized and will make contributions towards synthesizing our knowledge towards implementing more effective interventions in the future. I note these very few suggestions to incorporate into the protocol to improve its clarity and completeness:

1. It is unclear to me whether observational studies (cross-sectional or longitudinal) with appropriate control groups and adjustment for confounding would be included or excluded.

Response: Our inclusion criteria for impact evaluations are randomised and non-randomised studies, with the latter inclusive of studies that can reasonably address selection bias through incorporation of baseline measures, control groups and/or allocation rules such as regression discontinuity design. Additionally, observational studies will be included if they are linked to randomised and non-randomised quasi-experimental evaluations that meet study inclusion criteria. The following words in italics have been added in the subsection, ‘Methods, Study design, Study inclusion criteria,’ to clarify this point: “We will also include studies which are linked to the included intervention impact evaluation studies, including process evaluations, qualitative studies and cost or cost-effectiveness studies, i.e. *“trial sibling studies.”*”

2. This sentence, p.8, line 14: “An independent studies on women’s groups was not linked to an impact evaluation...” is unclear

Response: This sentence has been clarified as follows, with added word shown here in italics: An independent study such as a process evaluation, qualitative study and cost or cost-effectiveness study on women’s groups that was not linked to a randomised or non-randomised quasi-experimental impact evaluation (*non-sibling studies*)...

3. Given the requirement of an economic component to the interventions, the authors should also consider abstracting articles from an economic journal database (EconLit:

<https://www.aeaweb.org/econlit/>), where at least some microfinance group literature is published.

Response: Thank you for the suggestion. We have included EconLit as an additional database for review and added mention of this in the Methods, subsection Information sources, Electronic databases as follows, with added words in italics: We will search the following seven electronic research databases: Pubmed, Scopus, Embase, Web of Science, PsycINFO, Social Sciences Citation Index (SSCI), and EconLit.

Reviewer: 2

Dr. Valerie Flax, RTI International

Comments to the Author:

The review described by the authors is important as funders move forward with support for integrated or layered programming. It would be helpful if the authors could explain in the manuscript how they think their review differs from or adds to the previous two reviews by Orton et al. and Lorenzetti et al.

Response: Thank you for your encouraging comments. We have added text that specifies how our review builds on reviews conducted by Orton et al and Lorenzetti et al, as well as South Asia-focused reviews, in two specific ways. Orton et al included studies that measured the health effects of joining a microfinance group (without layering) as well as “layered” interventions, while Lorenzetti et al included all types of microfinance institutions, including non-group interventions. Our review builds on these important contributions by focusing only on integrated programming, in line with the increasing interest in group-based interventions that layer on additional programmes, such as India’s National Rural Livelihoods Mission and the Nigeria for Women Project. The added text reads: “We build on previous reviews by (i) focusing on the combined effect of group-based economic and health programs, as distinct from previous reviews that examined the health impacts of joining a microfinance-based group with or without layering⁴ or non-group-based microfinance interventions⁵ and (ii) including studies across all LMICs rather than a specific region 10 11.”

I would also like to offer the following minor comments for the authors’ consideration:

Intro, page 5 – first research question – It’s not clear what the authors mean by “single-purpose interventions”. Does that mean only an economic intervention without any layering? The authors may want to consider modifying the wording for clarity.

Response: We have modified this sentence as follows, with added words in italics: ‘1) What is the effect of combined economic and health, nutrition and/or sanitation interventions delivered through women’s groups on health outcomes amongst women and children in LMICs, compared to single-purpose, non-layered (i.e., only economic or only health) interventions with women’s groups or to no group intervention,

Page 6, lines 9-12 – Points 3 and 4 here seem contradictory. Does point 3 only refer to randomized studies?

Response: Thank you for this question; we clarified this by deleting point 4 and adding text as shown in italics: “Studies with the following characteristics will be excluded: 1) No qualitative or quantitative data were collected to evaluate the effect of an intervention, for example the study only provides observational descriptions of groups; 2) The treatment group only implements an economic or health intervention, i.e. the intervention is not an integrated approach; 3) No comparison group or baseline measurement was included, 4) An independent study such as a process evaluation, qualitative study and cost or cost-effectiveness study on women’s groups was not linked to a randomised or non-randomised impact evaluation (non-sibling studies); and 5) The study was published prior to January 1, 2000.

Page 6, line 28 – It may be hard to determine what percentage of the participants in a group were women in the case of mixed groups. Not all authors include the exact percentages of female and male group members. They may say “mostly women” or “a majority of the group members were women”. What will you do in such cases?

Response:

Thank you for pointing this out. We will include studies of mixed groups where the authors note that most of the members were women. We have edited the text as follows, with added text in italics: “Mixed-sex groups will be included if there were 51% or more women or authors note the group was composed mostly of women.”

Page 8, other sources – The authors mention earlier in the manuscript that they will search the grey literature, but in the “other sources” section there is no mention of grey literature searches. Will the authors search or contact some of the main NGOs that implement financial inclusion programs – CARE, Catholic Relief Services, Grameen Foundation, etc. – to obtain grey literature on combined/integrated programs?

Response: We will not systematically search databases that cover grey literature. We have deleted mention of this in the manuscript. We will include non-peer-reviewed published studies or reports in the grey literature identified through the following ways: We will compile a list of investigators of studies that we identify through personal knowledge or through our literature searches which appear to have the potential for an impact evaluation – for example, qualitative studies, study protocols, or studies which report designs that may qualify for inclusion in our review but do not include results of an evaluation in the manuscript and are not identified through database searches. We will reach out personally to the authors to explore whether an impact evaluation is available that is not yet published in the peer-reviewed literature. Further, we will consult with expert practitioners and researchers in this domain to identify missing studies in both the grey and peer-reviewed published literature. We will also search repositories of impact evaluations (3ie) and the Evidence Consortium on Women’s Groups.

We added the text below in italics to the subsection on Other sources: “Studies will also be obtained for the same time period from reference lists of included studies, reference lists of systematic reviews on the effects of women’s groups, the 3ie Database of Impact Evaluations and Evidence Consortium on Women’s Groups repository, and Google Scholar. In addition, we will contact study authors to identify process evaluations or qualitative studies conducted in conjunction with an impact evaluation and consult with experts familiar with interventions in Francophone Africa and Latin/South America to recommend inclusion of French or Spanish language articles. We will also contact study authors of published process evaluations or qualitative studies to identify linked, unpublished impact evaluations. Lastly, we will consult with experts in this area, both researchers and practitioners, to review our list of included studies to identify any interventions that may have been excluded.

VERSION 2 – REVIEW

REVIEWER	Rosenberg, Molly Indiana University School of Public Health-Bloomington, Epidemiology and Biostatistics
REVIEW RETURNED	24-Mar-2021
GENERAL COMMENTS	The authors have sufficiently addressed my previous comments. No further suggestions from me!